# Genome-Wide Identification, Characterization, and Expression Analysis of the Copper-Containing Amine Oxidase Gene Family in Mangrove *Kandelia obovata*

**DOI:** 10.3390/ijms242417312

**Published:** 2023-12-09

**Authors:** Quaid Hussain, Ting Ye, Chenjing Shang, Sihui Li, Jackson Nkoh Nkoh, Wenyi Li, Zhangli Hu

**Affiliations:** 1Shenzhen Engineering Laboratory for Marine Algal Biotechnology, Shenzhen Public Service Platform for Collaborative Innovation of Marine Algae Industry, Guangdong Engineering Research Center for Marine Algal Biotechnology, College of Life Science and Oceanography, Shenzhen University, Shenzhen, 518060, China; quaid_hussain@yahoo.com (Q.H.); 2100252001@email.szu.edu.cn (T.Y.); 18138206919@163.com (S.L.); nkohjackson@szu.edu.cn (J.N.N.); huzl@szu.edu.cn (Z.H.); 2College of Physics and Optoelectronic Engineering, Shenzhen University, Shenzhen 518060, China; 3Department of Biochemistry and Chemistry, La Trobe Institute for Molecular Science, La Trobe University, Bundoora, VIC 3086, Australia; wenyi.li@latrobe.edu.au

**Keywords:** *Kandelia obovata*, *CuAO* gene family, copper stress, phylogenetic analysis, gene expression profiling

## Abstract

Copper-containing amine oxidases (*CuAOs*) are known to have significant involvement in the process of polyamine catabolism, as well as serving crucial functions in plant development and response to abiotic stress. A genome-wide investigation of the CuAO protein family was previously carried out in sweet orange (*Citrus sinensis*) and sweet cherry (*Prunus avium* L.). Six *CuAO* (*KoCuAO1-KoCuAO6*) genes were discovered for the first time in the *Kandelia obovata* (Ko) genome through a genome-wide analysis conducted to better understand the key roles of the *CuAO* gene family in *Kandelia obovata*. This study encompassed an investigation into various aspects of gene analysis, including gene characterization and identification, subcellular localization, chromosomal distributions, phylogenetic tree analysis, gene structure analysis, motif analysis, duplication analysis, cis-regulatory element identification, domain and 3D structural variation analysis, as well as expression profiling in leaves under five different treatments of copper (CuCl_2_). Phylogenetic analysis suggests that these *KoCuAOs*, like sweet cherry, may be subdivided into three subgroups. Examining the chromosomal location revealed an unequal distribution of the *KoCuAO* genes across four out of the 18 chromosomes in *Kandelia obovata*. Six *KoCuAO* genes have coding regions with 106 and 159 amino acids and exons with 4 and 12 amino acids. Additionally, we discovered that the 2.5 kb upstream promoter region of the *KoCuAOs* predicted many cis elements linked to phytohormones and stress responses. According to the expression investigations, CuCl_2_ treatments caused up- and downregulation of all six genes. In conclusion, our work provides a comprehensive overview of the expression pattern and functional variety of the *Kandelia obovata CuAO* gene family, which will facilitate future functional characterization of each *KoCuAO* gene.

## 1. Introduction

Polyamines (PAs) play a crucial role in various aspects of plant development as well as in their responses to both abiotic and biotic stressors. PAs are organic cations with a low molecular weight that are widely dispersed in various living organisms [1]. According to Galston and Kaur-Sawhney [2], putrescine (Put), spermidine (Spd), and spermine (Spm) are the three predominant polyamines found in higher plants. The catabolism of PA is a crucial factor in the maintenance of PA homeostasis. Two distinct categories of amine oxidases play a role in the catabolism of polyamines: FAD-binding polyamine oxidase (PAO) and Copper amine oxidases (*CuAOs*) [3]. *CuAOs* play a crucial role in the cellular regulation of polyamine levels. Copper (II) amine oxidases catalyze the oxidation of primary amines, resulting in the formation of their corresponding aldehydes and hydrogen peroxide. Aldehydes serve as intermediates in diverse biosynthetic routes of alkaloids within the plant kingdom. *CuAOs* are believed to selectively oxidize polyamines in a single main amino group, leading to the formation of monocyclic structures in many instances. The involvement of these oxidases in the production of pyrrolizidine alkaloids has been hypothesized [1]. The *CuAO* enzyme in plants can be categorized into two distinct groups based on their subcellular localization. The first group is found within peroxisomes, as shown by the presence of peroxisomal targeting signals. The second group is located outside of the cell, as indicated by the presence of N-terminal signal peptides [4,5]. Previous studies have demonstrated the presence of *CuAO* members in both peroxisomal and extracellular locations across many species, such as Arabidopsis [6], apple [7], and sweet orange [5]. In the majority of instances, the catabolism of *Put* is facilitated by *CuAO* [3]. However, in Arabidopsis, both Put and Spd can be catalyzed by *CuAO* [4,6]. The homodimeric nature of plant *CuAOs* is characterized by the presence of a copper ion and a 2, 4, 5-trihydroxyphenylalanine quinone cofactor (TPQ) site within each subunit [8,9]. Nevertheless, it has been revealed by Planas-Portell et al. [6] that the *CuAOs* of Arabidopsis (*AtCuAO1–AtCuAO3*) are capable of utilizing both Put and Spd as substrates.

In tropical and subtropical coastal wetlands, mangroves constitute the main kind of halophytic vegetation and are important from an ecological standpoint. Furthermore, these organisms have demonstrated a high level of adaptation to the demanding conditions found in intertidal environments [10,11]. Throughout East Asian and Southeast Asian tropical and subtropical climates, salt marshes are the primary habitat for a woody plant known as *Kandelia obovata* [12,13,14]. *Kandelia obovata* is able to adapt to transitional habitats where land and ocean meet by surviving cyclic and aperiodic tidal impacts, which cause high salinity, severe erosion, and anaerobic conditions [15]. The preservation of biodiversity and mitigation of erosion are contingent upon the vital role played by *Kandelia obovata* [16,17]. The plant species *Kandelia obovata* in China exhibit variations based on the age of the mangrove forest in which they are farmed and consumed. Since the 1960s, a number of mangrove plantings, predominantly consisting of the *Kandelia obovata* species, have been effectively implemented with the aim of protecting the coastal environment. Consequently, mangrove forests of different ages containing *Kandelia obovata* have been planted [18]. 

Based on the findings of Wang et al. [19], it has been observed that *Kandelia obovata* mangroves exhibit a higher level of resilience towards waterlogging and display distinct responses to varying levels of light irradiance. Copper (Cu), lead (Pb), and zinc (Zn) are widely recognized as the predominant heavy metal contaminants, with mangroves playing a crucial role in their accumulation [20,21,22]. The growth of *Kandelia obovata* seedlings was found to be reduced with exposure to Cu [23,24,25]. The co-occurrence of multiple heavy metal stressors can lead to a notable degradation of chlorophyll in the leaves of *Kandelia obovata* [26]. Shen et al. [25] documented a reduction in the total chlorophyll concentration of *Kandelia obovata* when exposed to a composite synthetic wastewater solution containing several heavy metals. According to the findings of Zhao and Zheng [24], an increase in the concentration of Cu resulted in initial growth and subsequent fall in the amount of soluble sugar in the roots of *Kandelia obovata* seedlings. Conversely, the quantity of soluble sugar in the leaves exhibited a decrease [25]. Nevertheless, the impact of stress caused by Cu on *CuAOs* has not received sufficient consideration, as the majority of studies conducted on this species have mostly focused on examining physiological and molecular responses.

All living organisms require Cu, which is also a necessary protein cofactor for a variety of physiological processes [27]. Nevertheless, elevated levels of Cu have the potential to induce pollution, resulting in a deceleration of plant growth and hindered developmental processes [28]. Cano-Gauci and Sarkar [29] have reported that the reactivity of Cu can lead to notable oxidative damage in cells, hindered root growth, difficulties in flowering, and defects in germination. In response to both shortages and excesses of this vital element, eukaryotic organisms have developed a mechanism to efficiently manage the acquisition and distribution of Cu [30]. Cu is linked to several physiological functions in plants, including photosynthesis, mitochondrial respiration, superoxide scavenging, ethylene sensing, and cell wall metabolism [31]. Insufficient levels of Cu in plants lead to a multitude of abnormal phenotypes, including diminished water transport, distorted young leaves, and impaired growth and reproductive development [32]. In the investigation conducted by Shen et al. [25], it was noted that this particular species exhibited variable levels of tolerance towards different heavy metals. Notably, it showed the ability to withstand Cu stress up to a concentration of 400 mg/L. Consequently, the study encompassed a range of Cu solution concentrations spanning from 0 to 400 mg/L.

The *CuAO* gene family has not been found in *Kandelia obovata* as far as the current state of knowledge is concerned. Hence, this investigation represents the initial occurrence of a comprehensive genome-wide research aimed at identifying the presence of *CuAO* genes inside the genome of *Kandelia obovata*. The primary objective of this study was to find and characterize the six *CuAO* genes, followed by an examination of their expression levels in relation to five distinct CuCl_2_ treatments. Bioinformatics approaches were employed to investigate several elements pertaining to the evolution of *CuAO* genes in *Kandelia obovata*, with the aim of attaining a more comprehensive knowledge. The aforementioned aspects encompassed the examination of gene structures, physicochemical characteristics, chromosomal arrangement, duplication analysis, conserved motifs, cis elements, phylogenetic connections, subcellular localization, and the expression patterns of *CuAO* homologs. The current work aims to examine the characterization and expression analysis of the *CuAO* family in *Kandelia obovata*. The objective of this study is to develop a theoretical framework for future investigations into the response of the *CuAO* family to CuCl_2_ treatment in *Kandelia obovata* plants.

## 2. Results

### 2.1. Identification of CuAO Family Members in Kandelia obovata

*Kandelia obovata* contains a total of six *CuAO* genes in its genome, a notably higher count compared to other plant species like *Arabidopsis thaliana*, *Prunus avium* L., *Malus domestica*, *Heliotropium indicum*, *Glycine max*, and *Citrous sinensis*. The *CuAO* family demonstrated a molecular weight range of 11.78 to 17.09, with a mean value of 14.71 kilodaltons (kDa). The protein *KoCuAO1* demonstrated the greatest isoelectric point (pI) value of 9.72, while *KoCuAO3* revealed the lowest pI value of 6.82. The isoelectric point (pI) of the *CuAO* family displayed a mean range of 6.82 to 9.72, with a mean value of 8.62. The hydrophobic properties of six *CuAOs* were illustrated by the variability in their grand average hydropathy index (GRAVY) values, which ranged from 0.309 to 0.805.

The determination of the subcellular location of CuAO proteins can facilitate a deeper understanding of their molecular functionality. Based on the subcellular localization prediction of CuAO proteins (Table 1), it is likely that six *CuAOs* were distributed throughout several locations. The *CuAO* family demonstrated a mean amino acid length ranging from 133 to 159. The identified *CuAO* genes in *Kandelia obovata* were mapped to the correct chromosomes using the MapGene2Chromosome (MG2C) web tool, which was used to determine their relative chromosomal locations. The chromosomal positions of the six *CuAO* genes, specifically *KoCuAO1*, *KoCuAO2*, *KoCuAO3*, *KoCuAO4*, *KoCuAO5*, and *KoCuAO6*, were determined to be located on chromosome 04 (Chr04), chromosome 09 (Chr09), chromosome 13 (Chr13), and chromosome 15 (Chr15) (Figure 1 and Table 1).

### 2.2. Diversity within the CuAO Family in Relation to Domain and Three-Dimensional Structure

After aligning all *KoCuAOs*, the NCBI’s BLAST search showed that 60–100% of their amino acid sequences are identical. Additionally, these proteins exhibit sequence identity and similarity, with black representing 100%, green 80%, and grey 60% (Figure 2). The discovery of *CuAOs* in *Kandelia obovata* was made through domain analysis (PF01179), which revealed their presence in three transmembrane domains (Figure 2). The protein structures of *KoCuAOs* were verified utilizing the SWISS-MODEL workspace and SOPMA/prabi tool, as illustrated in Figure 3. The KoCuAO proteins were accurately simulated using the templates 5m94.1.A, as depicted in Figure 3. The range of sequence identity observed in this study varied from 71.3% to 81.66%. The values of GMQE ranged from 0.94 to 0.92, while the sequence coverage ranged from 0.93 to 1.00. The data presented in this study indicate that the 3D model predictions for KoCuAO proteins exhibit a high level of accuracy and demonstrate the presence of helix and strand structures. The CuAO proteins found in *Kandelia obovata* demonstrate secondary structures that are comparable to those observed in CuAO proteins of other species, as illustrated in Figure 4 and Appendix A.

### 2.3. CuAO Protein Phylogenetic Relationships

An unrooted neighbor-joining (NJ) tree was generated by aligning a total of 43 *CuAO* sequences from various plant species, including six from *Kandelia obovata* (Ko), 10 from *Arabidopsis thaliana* (At), four from *Prunus avium* L. (Pav), nine from *Malus domestica* (Md), five from *Heliotropium indicum* (Hi), one from *Glycine max* (Gm), and eight from *Citrous sinensis* (Cs). This facilitated the determination of the evolutionary relationships among the CuAO proteins derived from the seven species under investigation. Based on the evolutionary tree, it is possible to identify three distinct groups (I, II, and III) of CuAO proteins. Group I had a total of 31 CuAO proteins, which were categorized into several groups based on their species-specific characteristics. This subgroup was composed of the following proteins: four KoCuAO proteins (KoCuAO1/2/3/4), five AtCuAO proteins (AtCuAO2/3/4/5/10), three PavCuAO proteins (PavCuAO1/2/3), seven MdCuAO proteins (MdCuAO1/3/4/5/6/7/8), four HiCuAO proteins (HiCuAO1/2/3/5), one GmCuAO protein (GmCuAO1), and seven CsCuAO proteins (CsCuAO1/3/6/4/5/7/8). Group II consisted of a collective of eight CuAO proteins, encompassing one KoCuAO protein (KoCuAO6), two AtCuAO proteins (AtCuAO7/9), one PavCuAO protein (PavCuAO4), two MdCuAO proteins (MdCuAO2/9), one HiCuAO protein (HiCuAO4), and one CsCuAO protein (CsCuAO2). Group III consisted of three CuAO proteins, including one KoCuAO protein (KoCuAO5) and two AtCuAO proteins (AtCuAO1/6). Therefore, it is evident from the data shown in Figure 5 that Group I exhibited a greater abundance of CuAO members in comparison to Group II and III. It is observed that the *KoCuAOs* exhibit a higher degree of similarity with their counterparts found in *Arabidopsis thaliana*, *Malus domestica*, and *Citrous sinensis*, compared to other plant species such as *Prunus avium* L., *Heliotropium indicum*, and *Glycine max*.

### 2.4. Investigation of the Genetic Structure and Conserved Features of CuAO Genes

The investigation focused on the examination of the exon–intron patterns of the *CuAO* genes to explore the gene expansion of the *Kandelia obovata* family. In order to further our understanding of the structural attributes of *CuAO* genes, we examined the exon–intron structures and conserved motifs, as depicted in Figure 6. The number of *CuAO* exons ranged from 4 to 12, while the number of introns ranged from 1 to 2 (Figure 6). The gene family known as *CuAOs* exhibits a diverse range of gene structures, with the majority of *CuAO* genes containing one to two UTR/introns. However, specific members of the *CuAO* gene family, namely *KoCuAO1*, *KoCuAO2*, and *KoCuAO3*, possess only one intron. The highest count of exons identified in *KoCuAO3* and *KoCuAO4* was 12 and 11, respectively. The results of this study revealed that a group of individuals with *CuAOs* shared a gene structure that exhibited a high degree of similarity to that of their evolutionary relatives.

Furthermore, the MEME web servers were employed to elucidate the conserved motifs of the *CuAO* genes. In addition, six genes encoding *KoCuAOs* were found to contain six conserved motifs, as depicted in Figure 6. Motifs one, two, three, and six are present in all *KoCuAO* proteins, as indicated by the prediction. *KoCuAO1/2/5/6* identified motif four and motif five. The results of this study indicate that there is a notable level of similarity in the gene structure and amino acid sequence across individuals belonging to the same subfamily of *KoCuAOs*.

### 2.5. Cis-Regulatory Elements in the Promoters of Six CuAO Genes

Upon reviewing the existing literature on cis elements, which have the potential to shed light on the mechanisms involved in regulating gene expression, we proceeded to examine the promoter sequences located 2500 base pairs upstream of *CuAO* genes. Regarding the *CuAO* genes, it is worth mentioning that the light-responsiveness gene demonstrates a significant abundance of cis-elements. Additionally, it was observed that the promoter sequences of *CuAOs1-6* genes exhibit variations in the presence of cis elements linked with phytohormones such as Abscisic acid (ABA), Gibberellin (GA), Auxin, Salicylic acid (SA), and Methyl jasmonate (MeJA). Furthermore, other cis elements associated with defense and stress responses, including low temperature (Low Temp), drought, Zein metabolism, and anaerobic responses, were also detected (Figure 7). The presence of diversity in the response components serves as evidence-based proof for the regulatory functions of *CuAO* genes in a diverse array of physiological and biological processes.

### 2.6. Duplication Analysis of CuAO Gene Family

Segmental and tandem duplication processes play a pivotal role in facilitating the formation of novel gene families within plant genomes. Improved comprehension of the duplication events related to the *CuAO* gene in *Kandelia obovata*; an investigation was undertaken to analyze both segmental and tandem duplications within the gene family of *KoCuAOs*. The chromosomal distributions of six *KoCuAO* genes were assessed. The results of this investigation suggest that there was a solitary occurrence of segmental duplication involving the gene pairs *KoCuAO3* and *KoCuAO4* on chromosomes A09 and A14. This observation is illustrated in Figure 8. Tandem repeats of paralogous genes were seen in *KoCuAO1* and *KoCuAO2*, as well as *KoCuAO5* and *KoCuAO6*. Each of these pairs of genes was found to be present as a single gene, regardless of the chromosome on which they were located (Table 2). The findings of this study have demonstrated the significance of duplication events in the proliferation of the *KoCuAO* family genes. 

In order to improve understanding of the evolutionary constraints affecting the *KoCuAO* gene family, a comprehensive analysis was undertaken to ascertain the Ka (non-synonymous substitution rate), Ks (synonymous substitution rate), and the Ka/Ks ratio for *Kandelia obovata*. The duplicated gene pairs of *KoCuAOs* exhibited Ka/Ks ratios of 0.86, 0.97, and 0.99, suggesting that the *CuAO* gene family in *Kandelia obovata* might have undergone selective pressures or a discriminatory load throughout their evolutionary history (Table 2).

### 2.7. Expression Analysis of KoCuAO Genes under Copper Treatment

The quantitative real-time polymerase chain reaction (qRT-PCR) technique was employed to do expression profiling of six genes, namely *KoCuAO1*, *KoCuAO2*, *KoCuAO3*, *KoCuAO4*, *KoCuAO5*, and *KoCuAO6*. The present analysis was conducted with the inclusion of five separate CuCl_2_ treatments, specifically referred to as Cu0, Cu50, Cu100, Cu200, and Cu400 mg/L, as illustrated in Figure 9. In the current study, it was observed that the expression levels of *KoCuAO2* were decreased when exposed to a CuCl_2_ treatment of Cu200; however, they were increased under CuCl_2_ treatments of Cu50, Cu100, and Cu400 in comparison to the control treatment of Cu0. The *KoCuAO5* gene exhibited a decrease in expression levels following exposure to the Cu50, Cu200, and Cu400 treatments, while it showed an increase in expression in response to the Cu100 treatment, as compared to the control group (Cu0). The expression levels of the other two genes, namely *KoCuAO1* and *KoCuAO4*, exhibited considerable upregulation under all CuCl_2_ stress conditions as compared to the control condition without CuCl_2_. There were no significant differences found in the expression level of *KoCuAO3* between the Cu100 and Cu200 conditions.

Similarly, *KoCuAO6* did not reveal any significant differences in expression level between the Cu400 and Cu0 conditions. The expression levels of the four genes, *KoCuAO1-4*, were found to be greater in the Cu100 treatment, but the expression levels of *KoCuAO5* and *KoCuAO6* were higher in the Cu200 treatment compared to the other CuCl_2_ treatments. The observation above can be ascribed to the augmented expression levels of *KoCuAOs* in the leaf under conditions of restricted CuCl_2_ supply. Nevertheless, in the presence of substantial Cu, the transcript levels of these genes exhibited a decrease when compared to both the Cu0 condition and other Cu levels.

## 3. Discussion

*Kandelia obovata*, a member of the Rhizophoraceae family, is a crucial arboreal species found in tropical and subtropical regions of East and Southeast Asia. It serves as a vital coastal shelterbelt and contributes to the overall ecosystem [12]. *Kandelia obovata* exhibits adaptive characteristics in transitional habitats characterized by the interface between land and water. This adaptation enables the species to effectively cope with both periodic and aperiodic tidal influences, resulting in the presence of high salinity, severe erosion, and anaerobic conditions [15]. The species *Kandelia obovata* plays a vital role in the preservation of biodiversity and the mitigation of erosion [16,17]. Heavy metals are a prominent class of anthropogenic toxic substances found in mangrove ecosystems [23,26]. Among them, copper (Cu), zinc (Zn), and lead (Pb) are often observed as contaminants [33,34]. Cu is a vital micronutrient for the survival of living things. The process of Cu absorption across the cell membrane plays a vital role in the regulation of Cu homeostasis. Multiple variants of transporter proteins have been documented to facilitate the absorption of Cu [27,30]. In our recently published study by Hussain et al. [30], we discovered that various treatments of copper had a significant impact on the width and length of the leaves of *Kandelia obovata* plants. However, there were no noticeable differences in the height of the plants between the four copper treatments (Cu50, Cu100, Cu200, and Cu400 mg/L) and the control group without any heavy metals (Cu0). These data support a previous study by Chai et al. [35], which hypothesized that *Kandelia obovata* would devote more energy to leaf growth in response to many heavy metal stressors. These results corroborate those of previous studies by Cheng et al. [23], who discovered that throughout the 120-day experiment, certain heavy metal stressors had no discernible effect on plant growth.

The process of polyamine catabolism is of extreme significance in the context of plant development and the response to stress. Currently, the predominant emphasis in research is placed on investigating the involvement of polyamine oxidases (PAOs) in the degradation of polyamines (PAs) [5,36]. In contrast, the understanding of *CuAOs* remains limited despite the existence of several studies that have investigated their roles in polyamine catabolism and stress response [3,6]. Nevertheless, the response of *CuAO* genes in *Kandelia obovata*, a common woody mangrove species, to Cu stress (CuCl_2_) remains largely unexplored. In the present study, a genome-wide analysis was employed to identify a contracted gene family known as *CuAOs*, consisting of six members, within the genome of *Kandelia obovata*. The number of *CuAO* genes identified in various plant species is lower, comparatively. For instance, *Arabidopsis thaliana* has 10 *CuAO* genes [6], *Malus domestica* has nine [7], *Citrus sinensis* has eight [5], *Heliotropium indicum* has five [1], *Prunus avium* L. has four [4], and *Glycine max* has only one [37]. Based on the findings of published investigations, it has been observed that various plant species possess a range of one to 10 *CuAO* genes. Our research outcome aligns with a prior investigation in this regard.

The comprehension of the subcellular location of *KoCuAO* is of most importance in elucidating its functional role. The localization of *KoCuAOs* in *Kandelia obovata* has been demonstrated to occur in the peroxisome, cytoplasm, and chloroplast. This finding aligns with a prior study conducted on apple, which investigated the localization of the *MdAO1* protein and found similarities [7]. Our findings align with a prior work conducted on Arabidopsis, where we observed similarities between our results and the localization of *AtCuAO2* and *AtCuAO3* in peroxisomes [6]. The presence of a PTS1 in *CsCuAO1* and *CsCuAO3*, as determined by amino acid sequence analysis, indicates that these proteins may be localized in peroxisomes [5]. This observation is supported by the fact that PTS1 is a critical motif for peroxisomal localization, as reported by Planas-Portell et al. [6]. Furthermore, the presence of an N-terminal signal peptide was detected in *CsCuAO4*, *CsCuAO5*, and *CsCuAO6*, suggesting their potential localization as extracellular proteins. A study conducted by Wang et al. [5] demonstrated that five *CsCuAOs*, specifically CsCuAO4–8, were found to be situated on the same chromosome and displayed identical transcriptional orientation. This observation suggests that these *CsCuAOs* may have originated from a gene duplication event.

The phylogenetic analysis demonstrated that the *CuAO* genes derived from *Kandelia obovata* and six other plant species, namely *Arabidopsis thaliana* (At), *Prunus avium* L. (Pav), *Malus domestica* (Md), *Heliotropium indicum* (Hi), *Glycine max* (Gm), and *Citrus sinensis* (Cs), were categorized into two primary groups. The present study involved the identification of six *KoCuAO* genes from the *Kandelia obovata* genome. A phylogenetic analysis was conducted, which indicated that *KoCuAOs* and sweet cherry (*PavCuAOs*) had the closest relationship [4]. The findings of this study indicate a potential evolutionary trend within the *KoCuAO* gene family. The allocation of CuAO proteins in *Kandelia obovata* was divided between the three categories based on the previously established classifications, specifically those pertaining to sweet cherries [4]. Tandemly duplicated genes have a higher likelihood of being conserved throughout the process of evolution when they are involved in the response to environmental stimuli [38,39]. The gene structure of *KoCuAO* was subjected to analysis, which resulted in the identification of three distinct patterns. The number of exons in the *CuAO* gene ranged from 4 to 12, while the number of introns ranged from 1 to 2. The gene structure pattern seen in the study, as mentioned earlier, aligns with the gene structure patterns found in sweet cherry and sweet orange, as reported by previous research [4,5]. The rise in the number of exons and introns can be attributed to alterations in gene structure resulting from evolutionary processes. It has been observed that genes possessing a lower number of introns tend to undergo editing and exit the nucleus at an earlier stage in a general context [40,41]. Cis-regulatory motifs and elements found in promoter sequences are utilized to obtain a deeper understanding of how *KoCuAO* genes respond to various environmental stimuli. Cis elements, such as phytohormones (auxin, salicylic acid, gibberellin), zein metabolism, stress (low temperature, drought, defense, and stress), and meristem and endosperm expression, were included in this group. Prior studies have demonstrated that cis elements have a role in enhancing plant stress responses. The outcome of our study aligns with a prior discovery indicating that promoter cis elements have a significant impact on the regulation of gene transcription in plants, particularly in the context of tissue-specific or stress-induced expression patterns, as observed in [4]. 

Multiple investigations have established that *CuAOs* are implicated in many physiological responses to heavy metal exposure. The primary objective of this study was to examine the impact of Cu stress on the expression of *KoCuAO*, a specific gene. The results revealed that the expression levels of *KoCuAO2* were found to be downregulated when exposed to a Cu concentration of 200 mg/L, while they were upregulated in response to Cu treatments of 50, 100, and 400 mg/L, as compared to the control group (Cu0) [30]. The findings of our study align with previous research conducted on sweet cherry and sweet orange [5], suggesting that *PavCuAOs* may have diverse functional roles across various tissues. The expression of *PavCuAO1* and *PavCuAO3* was found to be significantly elevated in flowers and young leaves [4]. Conversely, *PavCuAO2* and *PavCuAO4* exhibited high levels of transcription in fruits, flowers, and young leaves. These findings imply that *PavCuAOs* may play a regulatory role in the development of several organs in sweet cherry [4]. The findings of our study are consistent with previous research conducted on the *CuAO* genes of Arabidopsis [6], sweet cherry [4], and sweet orange [5]. These collective findings indicate that the *CuAO* genes might play a role in the regulation of tissue development. The *KoCuAO* genes have the potential to serve as valuable genetic modifiers in enhancing the tolerance of crops and plants toward elevated levels of CuCl_2_.

## 4. Materials and Methods

### 4.1. Characterization and Identification of the CuAO Genes in Kandelia obovata 

The genome sequences of *Kandelia obovata* were acquired from the NCBI database (https://www.ncbi.nlm.nih.gov/; accessed on 15 October 2023, BioProject/GWH, accession codes: PRJCA002330/GWHACBH00000000) and the *Kandelia obovata* protein database (https://www.omicsclass.com/article/310; accessed on 15 October 2023) [12]. Two databases confirm hypothetical proteins: Pfam: http://pfam.xfam.org/ (accessed on 15 October 2023) and NCBI CDD (https://www.omicsclass.com/article/310, with an E-value of 1.2 × 10^−28^). The protein sequence analysis of copper amine oxidases (*CuAOs*) associated with the domain profile was performed using the Pfam database (http://pfam.xfam.org). The researchers utilized the *Kandelia obovata* genome database (https://www.omicsclass.com/article/310) and the National Center for Biotechnology Information (NCBI) database: https://www.ncbi.nlm.nih.gov/ (accessed on 15 October 2023) to ascertain and authenticate six *CuAO* family genes, specifically identified as *KoCuAO1*, *KoCuAO2*, *KoCuAO3*, *KoCuAO4*, *KoCuAO5*, and *KoCuAO6*. The Protparam tool, available at http://web.expasy.org/protparam/ (accessed on 15 October 2023), was employed to acquire physicochemical properties [42]. 

### 4.2. Chromosomal Distribution

Using the NCBI database and the web resource https://www.omicsclass.com/article/310, both accessed on 15 October 2023, the chromosomal locations and protein sequences of all *CuAO* genes in *Kandelia obovata* were discovered. The chromosomal locations of *CuAO* genes were also assessed on 15 October 2023. The researchers employed the MapGene2Chromosome (MG2C) method to ascertain the chromosomal positions of *CuAO* genes in *Kandelia obovata*. The MG2C utility, namely version 2.0, was accessed on 15 October 2023, via the following URL: http://mg2c.iask.in/mg2c [43].

### 4.3. Phylogenetic Tree Construction

The protein sequences of CuAO genes from the species *Kandelia obovata* (Ko), *Arabidopsis thaliana* (At), *Prunus avium* L. (Pav), *Malus domestica* (Md), *Heliotropium indicum* (Hi), *Glycine max* (Gm), and *Citrus sinensis* (Cs) were utilized in the phylogenetic analysis. The MEGA11 (V 6.06) software, which can be accessed at www.megasoftware.net (accessed on 15 October 2023), was frequently utilized for the purpose of aligning protein sequences [44]. The construction of the phylogenetic tree was performed utilizing the neighbor-joining (NJ) method, employing 1000 bootstrap iterations. The phylogenetic tree was analyzed and edited using Evolview (https://evolgenius.info//evolview-v2/#login) on 15 October 2023 [45].

### 4.4. Gene Structure and Significant Motif Analyses 

The genomic composition of *Kandelia obovata* has been elucidated, revealing the presence of six genes that are classified within the *CuAO* gene family. The exon/intron arrangements of the six *CuAO* genes were presented, along with the structural studies of the genes, using web software (http://gsds.cbi.pku.edu.cn, accessed on 15 October 2023) [45]. The web program MEME v5.4.1 was utilized on 15 October 2023, to identify further conserved strings or regions within the protein sequences of the six CuAOs proteins. This information can be accessible at https://meme-suite.org/meme/tools/glam2scan (accessed on 15 October 2023). The application utilized the subsequent configurations: alphabet sequencing including DNA, RNA, or protein; site distribution restricted to either zero or one occurrence per sequence (zoops); motif discovery mode established as classic mode; and a total of 10 motifs. The TBtools software (v1.106, http://www.tbtools.com/) was utilized to visualize the MEME results after obtaining the corresponding mast file [46].

### 4.5. Duplication Analysis

The analysis and visualization of the presence and duplication events of the *CuAO* family members in mangrove species were conducted using MCScanX: http://chibba.pgml.uga.edu/mcscan2/ (accessed on 15 October 2023), a software tool developed by the University of Georgia in Athens, GA, USA. The source was accessed online on 15 October 2023. The present study employed Circos visualization to identify duplications in the *Kandelia obovata* species. Furthermore, MCscanX was utilized to conduct a comprehensive search for *CuAO* genes in Ko. The KaKs Calculator 2.0 (https://sourceforge.net/projects/kakscalculator2/, accessed on 15 October 2023) was utilized to calculate the synonymous (Ks), non-synonymous (Ka), and Ka/Ks ratios for the purpose of investigating the evolutionary constraints of the *CuAO* gene pairs [47].

### 4.6. Cis-Regulatory Elements (CREs)

The upstream sequences, including 2500 base pairs, of the *CuAO* family members were collected from the genome assembly database of *Kandelia obovata*. The PlantCARE tool, accessible at http://bioinformatics.psb.ugent.be/webtools/plantcare/html/ (accessed on 15 October 2023), was utilized for the identification of CREs within the acquired sequences. The generation of Figure 6 in TBtools (v1.106) involved the utilization of the most commonly occurring CREs that were identified for the *CuAO* genes. This was accomplished through a thorough investigation of the frequency of each CRE motif [30].

### 4.7. Three-Dimensional Structure and Subcellular Localization

The estimation of the three-dimensional (3D) structure can be conducted by the utilization of SWISS-MODEL, an online resource accessible at https://swissmodel.expasy.org/interactive, which was accessed on 15 October 2023 [45]. In this study, the TMHMM-2.0 tool: https://services.healthtech.dtu.dk/service.php?TMHMM-2.0 (accessed on 15 October 2023) was employed for the prediction of transmembrane helices in proteins. Additionally, the secondary structure was determined using the SOPMA/prabi method: https://npsa-prabi.ibcp.fr/cgi-bin/npsa_automat.pl?page=npsa_sopma.html (accessed on 15 October 2023) [38]. Two online techniques were utilized to estimate the subcellular localization of the *CuAO* family genes.

ProtComp 9.0: http://linux1.softberry.com/berry.phtml?topic=protcomppl&group=programs&subgroup=proloc (accessed on 15 October 2023), is a software application utilized for the purpose of analysis. The CELLO server located at cello.life.nctu.edu.tw was accessed on 15 October 2023 [30].

### 4.8. Plant Material and Environmental Conditions

The study utilized seedlings of *Kandelia obovata* that were one year old and used three plots per treatment, 10–12 plants per plot, and three replications. These seedlings were planted in the mangrove conservation site located in Golden Bay Mangrove Reserve, situated in Beihai, Guangxi Province. The geographical coordinates of the site are 109.22° N and 21.42° E. The soil was irrigated with CuCl_2_ irrigation and received daily morning and evening watering with seawater from the adjacent vicinity, as a component of semi-natural agriculture practices. Throughout a two-year period, five distinct concentrations of CuCl_2_ were employed for the treatments, namely Cu0, Cu50, Cu100, Cu200, and Cu400, corresponding to concentrations of 0, 50, 100, 200, and 400 mg/L, respectively. The control treatment utilized local seawater as the starting concentration, with a Cu0 content of 0 mg/L. Different CuCl_2_ concentrations (Cu0, Cu50, Cu100, Cu200, and Cu400 mg/L) affect leaf morphology and plant growth, and their photographs depicting which are already discussed in our recently published article by Hussain et al. [30]. The soil sample used for this experiment had a background Cu concentration of <1.0%, which was classified as non-polluted [48]. Following a two-year period of treatment, plant samples were collected in order to assess the various criteria [30].

### 4.9. Quantitative Real-Time PCR Assays 

The extraction of total RNA from the aforementioned leaves was performed using TRIzol reagent (Invitrogen, Carlsbad, CA, USA, http://www.invitrogen.com; accessed on 15 October 2023). The ABI PRISM 7500 Real-time PCR Systems, manufactured by Applied Biosystems in Waltham, MA, USA, were employed to conduct quantitative real-time PCR (qRT-PCR) assays. The 2^−∆∆CT^ approach, as previously outlined by Hussain et al. [30] was utilized for these experiments. The reference gene for *Kandelia obovata* (KoActin) was selected based on the sequence supplied by Sun et al. [13]. The forward primer used was CAATGCAGCAGTTGAAGGAA, and the reverse primer used was CTGCTGGAAGGAACCAAGAG. The exact *KoCuAO* gene primers utilized for real-time PCR are presented in Table 3. These primers were produced using the primer-designing tool provided by the National Center for Biotechnology Information (NCBI) (https://www.ncbi.nlm.nih.gov/tools/primer-blast/, accessed on 8 April 2023).

### 4.10. Statistical Analysis

Statistix 8.1, an analytical program created by a Tallahassee, Florida, USA-based Corporation, was utilized to analyze the data. The analysis utilized a one-way analysis of variance (ANOVA), and the findings were reported in terms of the mean and standard deviation (SD) of the three biological replicates. The primary objective of this study was to investigate the variations in leaf structure among plants exposed to five distinct levels of copper stress (Cu0, Cu50, Cu100, Cu200, and Cu400 mg/L), which are already discussed in our recently published article by Hussain et al. [30]. A least significant difference (LSD) test was conducted at a significance level of *p* < 0.05. The graphs were produced using GraphPad Prism version 9.0.0, a statistical software for Windows developed by GraphPad Software in San Diego, California, USA (https://www.graphpad.com; accessed on 25 October 2023).

## 5. Conclusions

A total of six putative *CuAO* genes (designated as *KoCuAO1-6*) were discovered in the species *Kandelia obovata*. The presence of *KoCuAO* genes was detected on the four chromosomes of *Kandelia obovata*. The expression patterns of *KoCuAO* genes exhibited variations within Ko leaves, suggesting that *KoCuAOs* are involved in tissue-specific processes. Based on the examination of gene expression in response to CuCl_2_ treatment, it was noted that a majority of genes exhibited considerable upregulation when the CuCl_2_ concentration was reduced. Conversely, when the CuCl_2_ concentration was increased, the expression level of *CuAO* genes was found to be downregulated. These findings will also facilitate the identification of potential genes that enhance plant architecture in response to stressful situations and enable potential genetic enhancements and breeding of *KoCuAO* genes in other crops. This may include techniques such as CRISPR/Cas-mediated deletion, overexpression, and other genetic modifications.

## Figures and Tables

**Figure 1 ijms-24-17312-f001:**
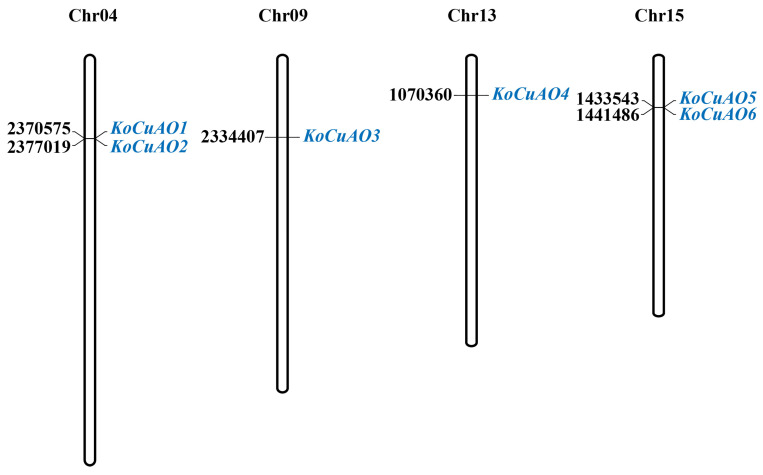
Illustrates the schematic depiction of the spatial arrangement of the *CuAO* gene throughout the four chromosomes of *Kandelia obovata*. The gene’s name is denoted in bule color on the right-hand side. The chromosomal locations in which the *CuAO* genes are indicated are represented by black letters. The numerical designations of chromosomes, known as chromosomal numbers (Chr), are typically located in the topmost area of each chromosome.

**Figure 2 ijms-24-17312-f002:**
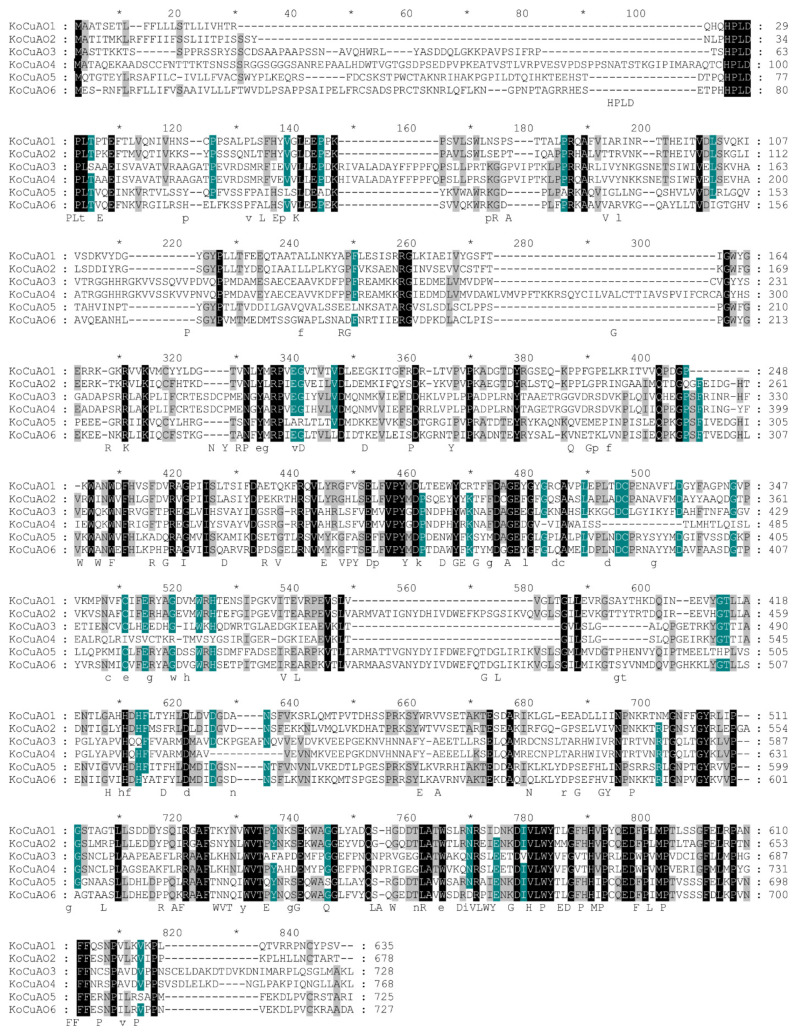
The amino acid sequence was subjected to several alignments using data obtained from each *KOCuAO* gene. Sequence identity and similarity were displayed as 100% in black text, 80% in green text, and 60% in grey text. The symbols “*” that appear above the sequence indicate every 10 residues of amino acids.

**Figure 3 ijms-24-17312-f003:**
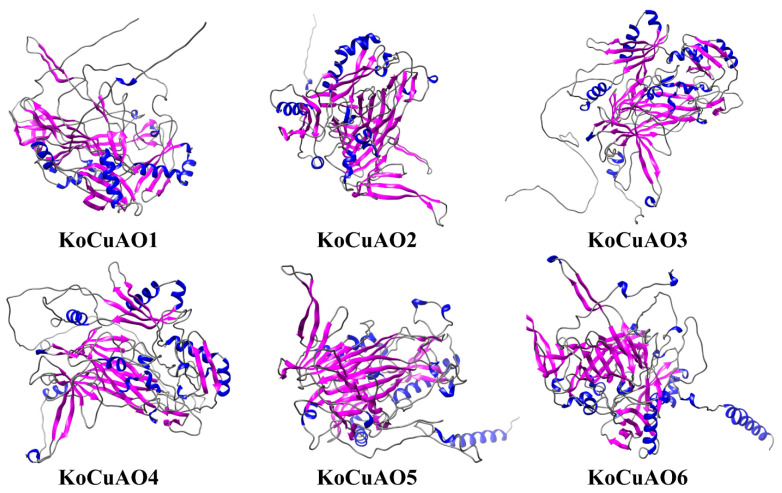
The three-dimensional (3D) structures of KoCuAO1-6. The SWISS-MODEL software, an online resource accessible at https://swissmodel.expasy.org/interactive, which was accessed on 15 October 2023, was employed for the prediction of three-dimensional structural homology models. The blue color characterizes the helix, whereas the pink color distinguishes the strand.

**Figure 4 ijms-24-17312-f004:**
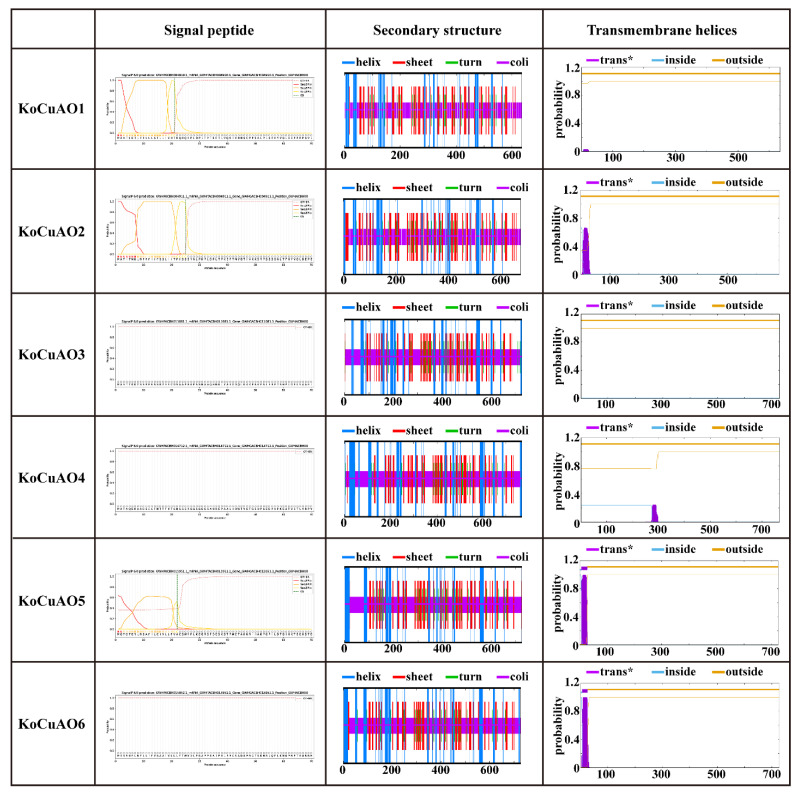
The transmembrane structures of the KoCuAO1-6 proteins. The validity of the transmembrane structures was confirmed through the utilization of the SOPMA/prabi programme. Trans* indicated the transmembrane.

**Figure 5 ijms-24-17312-f005:**
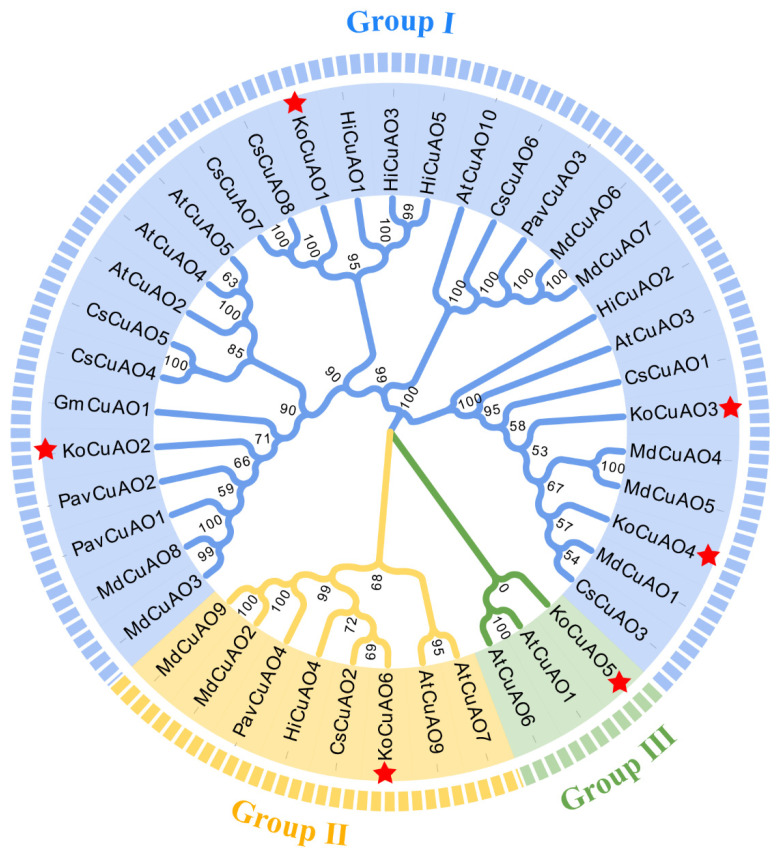
The phylogenetic study of CuAO proteins from *Kandelia obovata* (Ko), *Arabidopsis thaliana* (At), *Prunus avium* L. (Pav), *Malus domestica* (Md), *Heliotropium indicum* (Hi), *Glycine max* (Gm), and *Citrous sinensis* (Cs) was conducted using the most significant likelihood technique. There are three distinct groups of CuAO proteins, namely Group I, Group II, and Group III, each characterized by a unique color. The presence of CuAO proteins in *Kandelia obovata* is indicated by the red color of the star symbol “*”.

**Figure 6 ijms-24-17312-f006:**
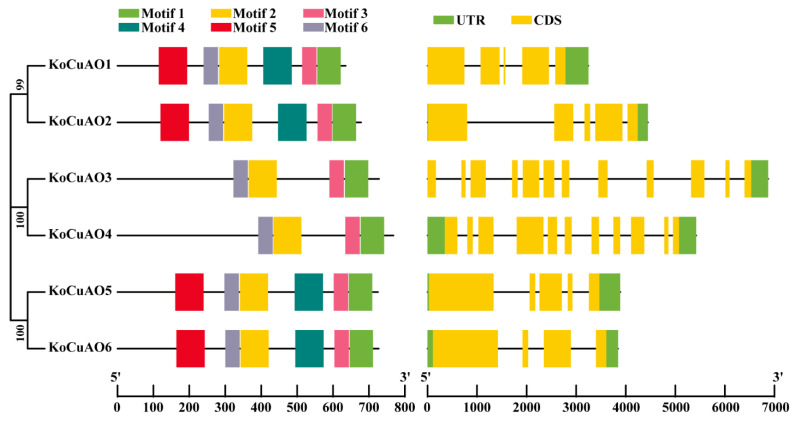
The gene structure and motif composition of the *CuAO* family genes in *Kandelia obovata* were investigated. The *CuAO* genes present in both genomes were categorized into three distinct groups based on their evolutionary relationships, specifically focusing on the gene structure of the *CuAOs*. The color green characterizes the visual representation of the UTR sections, whereas the CDS or exons are shown in yellow. The presence of introns is indicated by a horizontal line that is colored black. Furthermore, the conserved motif structures that have been found in the *CuAOs* are designated by a specific letter. Different colored boxes exhibit unique motifs.

**Figure 7 ijms-24-17312-f007:**
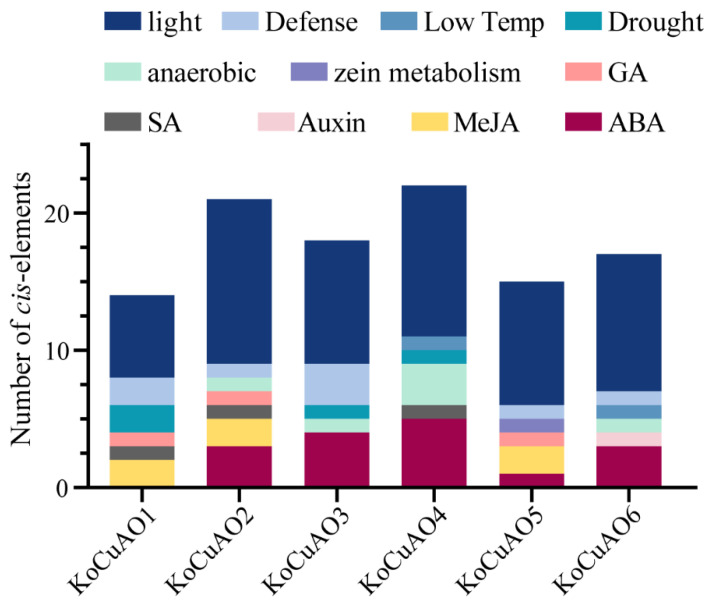
Regulatory elements, commonly known as CREs, have been identified inside the promoters of the *KoCuAO* gene. The vertical bars illustrate the positional distribution of the predicted CREs on the promoters of *KoCuAOs*. The promoter sequences of six *KoCuAO* genes, each spanning 2500 base pairs, were subjected to analysis using the PlantCARE tool. Various colors were used in this legend to represent different cis elements symbolically.

**Figure 8 ijms-24-17312-f008:**
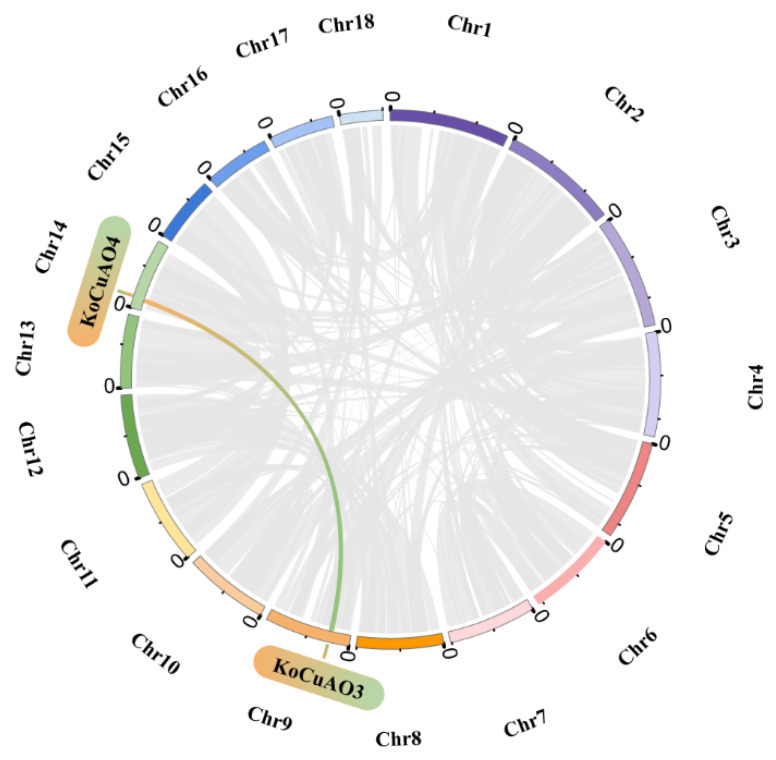
Circles serve as a visual representation of the distribution of the *KoCuAO* gene chromosomes and the interchromosomal connections. The orange and green lines represent the gene pair of *CuAOs*, while the syntenic blocks in the genome of *Kandelia obovata* are depicted by the grey lines in the background. Chromosomes 1–18 are shown with different colors and in a circular form.

**Figure 9 ijms-24-17312-f009:**
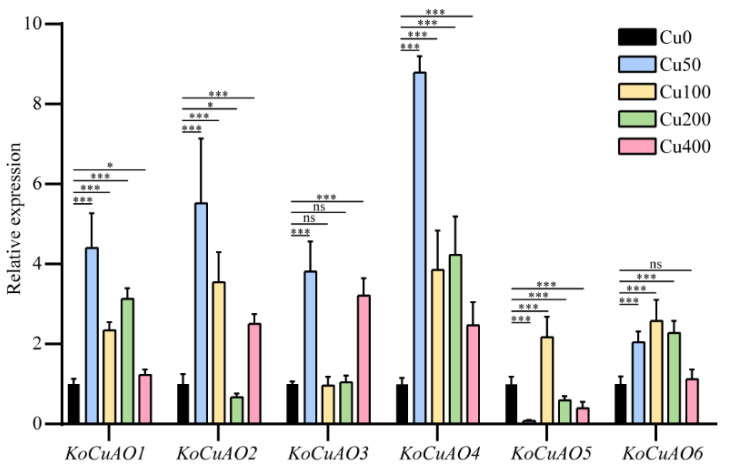
Expression of *KoCuAOs* in the leaves of seedling-stage *Kandelia obovata* plants under Cu0, Cu50, Cu100, Cu200, and Cu400 mg/L CuCl_2_ stress conditions was found using qRT-PCR analysis. There is a statistically significant difference (*p* < 0.05) observed between the control group and all other circumstances, as determined by the least significant difference (LSD) test. The presence of an asterisk (*) indicates the existence of notable disparities, with * denoting a significance level of *p* < 0.05 and *** indicating a significance level of *p* < 0.001. The “ns” shows non-significant differences. The relative gene expression is represented on the vertical axis, while the *CuAO1–6* genes are depicted on the horizontal axis.

**Table 1 ijms-24-17312-t001:** Provides a detailed overview of the *CuAO* gene family identified in *Kandelia obovata*.

Name	Gene ID	Location	AA ^1^	Chains ^2^	MW ^3^/kDd	pI ^4^	GRAVY ^5^	Subcellular (WoLF PSORT)	Subcellular (Plant-Ploc)
*KoCuAO1*	geneMaker00012803	Chr04 2370575-2373822	159	+	17.09	9.72	0.309	Chloroplast	Cytoplasm
*KoCuAO2*	geneMaker00012751	Chr04 2377019-2381467	106	-	11.18	8.71	0.498	Cytoskeleton	Cytoplasm
*KoCuAO3*	geneMaker00008448	Chr09 2334407-2341281	145	-	15.64	6.82	0.566	Peroxisome	Chloroplast
*KoCuAO4*	geneMaker00006577	Chr13 1070360-1075782	133	-	14.78	8.82	0.805	Peroxisome	Chloroplast
*KoCuAO5*	geneMaker00006577	Chr15 1433543-1437433	133	-	14.78	8.82	0.805	Peroxisome	Chloroplast
*KoCuAO6*	geneMaker00006577	Chr15 1441486-1445331	133	-	14.78	8.82	0.805	Vacuole	Chloroplast

AA ^1^: Number of amino acids; Chains ^2^: Positive or negative chains; MW ^3^: Molecular weight; pI ^4^: Isoelectric point; GRAVY ^5^: Grand average of hydropathicity; Ko: *Kandelia obovata*; Chr: Chromosome

**Table 2 ijms-24-17312-t002:** Comprehensive data regarding the Ka, Ks, and Ka/Ks ratio in *Kandelia obovata*.

Name	Method	*Ka*	*Ks*	*Ka/Ks*	Divergence-Time (MYA)	Duplicated Type
*KoCuAO1* and *KoCuAO2*	MS	0.97	1.13	0.86	37.56	tandem
*KoCuAO5* and *KoCuAO6*	MS	0.99	1.03	0.97	34.17	tandem
*KoCuAO3* and *KoCuAO4*	MS	1.00	1.01	0.99	33.67	segmental

MYA: million years ago.

**Table 3 ijms-24-17312-t003:** Lists the primers utilized in this study’s qRT-PCR gene expression investigation.

Gene Name	Primer Name	Sequence (5’-3’)	Length	Tm	GC%	Product Length
*KoCuAO1*	1-F	GTCAACCACAGCCCTACCTC	20	60.04	60	165
	1-R	GTGGCTGCTGTTTGCTCTTC	20	60.04	55	
*KoCuAO2*	2-F	TCAGAACCCACCATCCAAGC	20	59.96	55	223
	2-R	TGCCTCTGTTTTCAGCCGAT	20	59.96	50	
*KoCuAO3*	3-F	TGTCGCATCTCAACGAAGCA	20	60.32	50	143
	3-F	CAAGTCCCCAAGGCTGCATA	20	60.03	55	
*KoCuAO4*	4-F	GTGTCTGTTTGCACGAAGAGG	22	59.4	54.5	193
	4-R	TTCACCTCAACAACTGCCAT	22	59.5	59.1	
*KoCuAO5*	5-F	CACTCGTTGTCACTGGACGA	20	59.97	55	92
	5-R	CCGATCACTTGGGCTTTCCT	20	60.04	55	
*KoCuAO6*	6-F	TCACACGGACTCGTTCACAG	20	59.97	55	157
	6-R	AGACAATCGCTGCACTCACA	20	59.97	50	

## Data Availability

Data are contained within the article or Appendix A.

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
