# Peer review of "Genome-Wide Identification, Characterization, and Expression Analysis of the Copper-Containing Amine Oxidase Gene Family in Mangrove Kandelia obovata"

_ijms, 2023, doi:10.3390/ijms242417312_

Round 1
Reviewer 1 Report
Comments and Suggestions for Authors
The manuscript "Genome-wide Identification, characterization, and expression analysis of copper-containing amine oxidases (CuAOs) gene family in mangrove Kandelia obovata" is good work. Methods are robust and results are well presented. Please address the following comments.
Make the scientific names of species and genes in italics throughout the manuscript. One example is line 28.
Clarify the controversy between line 143-144 and line 431-434. Tools are not matching.
Line 152: out of the five chromosomes of Kandelia obovate?
Line 152: italic the scientific name
Italic the gene names on a map
Line 186-205: What is the conclusion of the evolutionary relationship?
Line 522: variations in average leaf values among plants? Could you elaborate on this a bit more?
Reviewer 2 Report
Comments and Suggestions for Authors
The photographs depicting plants treated with different copper (Cu) concentrations must be provided.
In line 497, it should be “CuCl2”.
Additionally, essential information on the methodology, particularly regarding growth conditions such as soil properties, the number of plots per treatment, the number of plants per plot, and the number of replicates, needs to be provided.
In line 521, it mentions the standard deviation (SD) of the tree replicates. It is not clear whether these are repetitions of the same plants or different ones.
After analyzing Figure 9, it appears that only a t-test was performed, indicating a single-group comparison. I suggest conducting a comprehensive one-way analysis of variance (ANOVA) with post-hoc analyses using Tukey’s test or Dunnett's test for comparison with the control group. However, it is important to check the homogeneity of variance (Levene's test) beforehand, as Figure 9 suggests a potential issue. If variance is not homogeneous, nonparametric tests may be necessary.
Consider combining the results and discussion sections and providing a more in-depth causal discussion of the factorial experimental results.
Round 2
Reviewer 2 Report
Comments and Suggestions for Authors
The paper can be published in the present form.